# Feasibility of a Virtual Educational Programme for Behaviour Change in Cardiac Patients from a Low-Resource Setting

**DOI:** 10.3390/ijerph20115934

**Published:** 2023-05-24

**Authors:** Rafaella Zulianello dos Santos, Sidnei Almeida, Andrea Korbes Scheafer, Marlus Karsten, Paul Oh, Magnus Benetti, Gabriela Lima de Melo Ghisi

**Affiliations:** 1Cardio Oncology and Exercise Medicine Nucleus (NCME), University of Santa Catarina State, Florianopolis 88080-350, Brazil; 2Physical Therapy Department, University of Santa Catarina State, Florianopolis 88080-350, Brazil; 3KITE Research Institute, University Health Network, Toronto, ON M4G 2V6, Canada

**Keywords:** cardiac rehabilitation, feasibility study, patient education as topic, virtual education

## Abstract

Patient education is an integral part of recovery from a critical cardiac life event and a core component of cardiac rehabilitation (CR) programmes. This study addressed the feasibility of a virtual educational programme for behaviour change in CR patients from a low-resource setting in Brazil. Cardiac patients from a CR programme closed due to the pandemic received a 12-week virtual educational intervention (WhatsApp messages and bi-weekly calls from healthcare providers). Acceptability, demand, implementation, practicality, and limited efficacy were tested. Overall, 34 patients and 8 healthcare providers agreed to participate. The intervention was considered practical and acceptable by the participants, who reported a satisfaction median of 9.0 (7.4–10.0)/10 (patients) and 9.8 (9.6–10.0)/10 (providers). The main difficulties in carrying out the intervention activities were related to technology, motivation to self-learning, and a lack of in-person orientation. All the patients reported that the information included in the intervention was aligned with their information needs. The intervention was associated with changes in exercise self-efficacy, sleep quality, depressive symptoms, and performance of high-intensity physical activity. In conclusion, the intervention was considered feasible to educate cardiac patients from a low-resource setting. It should be replicated and expanded to support patients that face barriers to onsite CR participation. Challenges related to technology and self-learning should be addressed.

## 1. Introduction

Cardiovascular diseases (CVDs) are the leading cause of disease burden worldwide [1], with the highest mortality and morbidity rates found in low and middle-income countries (LMICs) [2]. The estimated prevalence of heart conditions in the Brazilian adult population is 32% [3]. Brazil is a large country with an evolving economy but marked social inequalities [4]; therefore, there is a great need for cost-effective cardiovascular secondary prevention, including cardiovascular rehabilitation (CR).

CR is an outpatient model of secondary preventive care proven to mitigate the CVD burden [5,6]. It involves the delivery of multiple core components, such as exercise training, patient assessment, risk factor management, and patient education and counselling [7]. Despite the well-established benefits, including a reduction in cardiovascular morbidity and mortality, an increase in quality of life, and cost-effectiveness [8], the availability and characteristics of CR programmes in Brazil are scarce, with 75 programmes identified via a 2018 global survey of programmes [9,10], and 500,000 more spots are needed per year to treat patients [10]. It was also identified through this survey that Brazilian CR programmes were not exploiting alternative delivery settings, such as home-based and virtual services [10].

When the coronavirus disease 2019 (COVID-19) was declared a pandemic in early 2020, the delivery of CR was impacted, with many programmes globally closing or switching to virtual delivery [11]. In Brazil, 31% of programmes started offering remote CR services [12]. In regard to specific CR core components, a significant decrease in the number of patient education interventions delivered was observed [11]. As the pandemic evolved, patients were fearful of their cardiac care [13], and healthcare providers were not trained or prepared to deliver their care virtually [11,14]. Despite the shift in the adoption and use of virtual care to reduce the risk of virus transmission, there is limited data that explores the feasibility of virtual programmes in the care of these patients, including the delivery of educational programmes to support behaviour change. In this context, the current study aimed to assess the feasibility of a virtual educational programme for behaviour change in CR patients from a low-resource setting.

## 2. Materials and Methods

### 2.1. Design

This was a longitudinal study designed to assess the feasibility—the acceptability, demand, implementation, practicality, and limited efficacy testing [15,16]—of a virtual educational programme for CR patients from a low-resource setting in Brazil. The study was approved by the Santa Catarina State University’s Ethics Board (40252720.3.0000.0118). The data were collected between May 2020 and February 2021.

### 2.2. Participants

The sample was recruited from a public CR programme in the south of Brazil (Cardio-Oncology and Exercise Medicine Center, Florianopolis, Brazil). During the study, this CR program was closed due to COVID-19 restrictions, and patients were not being assisted or contacted by the healthcare team. The inclusion criteria for the patient participants were the following: a confirmed cardiac diagnosis or presence of cardiovascular risk factors, being able and willing to provide informed consent, and having internet access. The exclusion criteria for the patient participants were the following: younger than 18 years old, illiterate, and any impairment that would preclude the participant’s ability to access educational materials and complete the questionnaires. All eight healthcare providers involved in the delivery of the educational intervention (all physiotherapists) also participated in this study and provided information on acceptability and implementation.

### 2.3. Procedures

The individuals who were eligible for the study were contacted by phone and received explanations about the study procedures, including a video with details about the educational intervention. Those who accepted signed the consent form. In addition, a phone interview was scheduled to complete pre-test assessments (15 days prior to the start of the intervention). During the call, the patient participants completed the assessments online via Google Docs, with the research team member being available for questions. Post-test assessments were also completed online by the patient participants, again with a research team member being available for questions (1 week after the end of the intervention). Within 6 months of the end of the intervention, the participants were again contacted by phone to report the acceptability, demand, and practicality of the intervention.

The healthcare providers involved in the delivery of the educational intervention participated in a training session about adult learning principles and patient education strategies. At the end of the intervention, they were invited to complete an online survey and attend one 1:1 semi-structured interview (with a research team member not involved in the delivery of the intervention) to collect data on acceptability and implementation.

### 2.4. Intervention

The education programme was based on the virtual Cardiac College™ curriculum, an evidence-based and theoretically informed comprehensive educational programme for CR [17]. It comprised 12 learning plans delivered weekly to the patient participants via WhatsApp in combination with 6 1:1 bi-weekly phone calls (15–30 min in duration). The learning plans included a review of the learning goals, direct links to the learning materials (including 12 short videos and 9 patient booklets) and 2 tools for self-management (action plans and reflection diaries). The content included education on exercise, diet, psychosocial health, medication, and action planning. During the phone calls, the patient participants were able to ask questions and the providers reinforced the important components of the intervention, including the completion of weekly action plans and reflection diaries. Figure 1 illustrates the educational intervention, including all the educational topics delivered.

### 2.5. Feasibility Measures

The feasibility was investigated in terms of acceptability, demand, implementation, practicality, and limited efficacy testing [15,16]. Table 1 describes these outcomes, along with the data sources and data analysis. Appendix A presents the questions asked to the patients and healthcare providers to assess these measures.

The following measurements (and corresponding psychometric-validated questionnaires in Brazilian-Portuguese) were used to investigate limited efficacy: disease-related knowledge measured by the short version of the Coronary Artery Disease Education Questionnaire (CADE-Q SV) [18], exercise self-efficacy measured by Bandura’s Exercise Self-Efficacy Scale (SES) [19], sleep quality measured by the Pittsburgh Sleep Quality Index (PSQI) [20], depressive symptoms measured by the short version of the Patient Health Questionnaire (PHQ-2) [21], and physical activity level measured by the International Physical Activity Questionnaire Short Form (IPAQ-SF) [22], which was chosen to assess this outcome due to the experience of the Brazilian research team with this tool.

### 2.6. Data Analysis

The sample size calculation followed the rule of a minimum of 30 patient participants to estimate the feasibility outcomes [23]. The patient participants were characterised according to age, gender, duration of CR participation, marital status, educational level, family income, and clinical diagnosis. The Shapiro–Wilk test was used to test the normality of the sociodemographic and clinical data distribution. The continuous variables with normal distributions were expressed as the mean and standard deviation, while those with non-normal distributions were expressed as the median and interquartile range.

The data were exported from Google Docs to SPSS version 20, where all the analysis was performed. Descriptive statistics (e.g., frequency with percentage) were applied for all the closed-ended items in the survey. All the open-ended responses were coded using an interpretive-descriptive approach. Additionally, the difference between the post- and pre-intervention scores (Δ post-pre) was calculated for limited efficacy testing, as previously described.

## 3. Results

### 3.1. Patient Participants

Overall, 104 patient participants were invited to participate in this study, of which 34 (33%) signed the consent form and received the 12-week virtual educational intervention. The sociodemographic and clinical characteristics of these participants are described in Table 2. The mean age was 63.8 ± 6.7 years, with the majority being male (n = 22; 65%), with an educational level equal to or lower than high school (n = 20; 59%), with a monthly salary equal to or lower than USD 1000 (n = 20; 59%), mainly with a diagnosis of heart failure (n = 22; 65%) or acute myocardial infarction (n = 21; 62%), and with hypertension (n = 25; 74%). In addition, 77% of the sample was composed of patients that were participating in the CR programme for 13 months or more.

### 3.2. Acceptability

Table 3 presents the results for the feasibility outcomes, which were reported by 20 patient participants (lost to follow-up = 14) and 8 healthcare providers. In regard to acceptability, the patient participants were highly satisfied with the virtual education, with all the median scores of satisfaction being higher than 9 out of 10. The healthcare providers were also highly satisfied with the educational intervention, overall, the content and delivery (median scores higher than 8/10). The top features of the programme highlighted by the patients were the education about exercise, diet, medication, and how the heart works. They also reported the least favourite features of the programme as not being able to learn in person and having to fill out action plans and diaries every week.

### 3.3. Demand

Regarding demand, most of the patient participants had internet access and used a cell phone to access the Internet (Table 3). In addition, these participants were already using the Internet to search for health information prior to the intervention. The patient participants reported that the main reasons to participate in the virtual educational programme were learning about exercise and overall health condition, maintaining or improving health through education, and staying in touch with the CR programme during the pandemic. Finally, the majority of the patients perceived the information received as useful and the bi-weekly calls as effective.

### 3.4. Implementation

Concerning implementation, access and the use of virtual education tools are reported in Table 3. Overall, all 34 patient participants accessed all 12 weekly educational sessions and watched the corresponding video and read parts of the booklet. Weekly learning plans were created by 18 patients, with weeks 1 (create a plan for change), 2 (maintain your aerobic exercise programme) and 9 (manage depression and stress) being the ones where the majority (i.e., >50%) of participants created a learning plan. In addition, reflection diaries were created by 16 patients, with again weeks 1, 2, and 9 being the ones where more than 50% of the participants completed this activity.

The healthcare providers’ perceptions about the success of the intervention included bi-weekly phone calls and educational content aligned with the needs of their patients. Their perceptions regarding failure included a lack of in-person contact, a lack of diabetes-specific content, the language of educational materials sometimes being difficult for some patients, and the educational content being delivered via message and not via an intuitive and patient-friendly digital platform.

### 3.5. Practicality

In regard to practicality, 95% of the patient participants reported that they changed their heart health behaviours after the intervention (Table 3). The main behaviour changes self-reported by the patients were “more exercise”, “better diet”, “being more optimistic about the future”, “better sleep”, and “better self-care”.

All the healthcare providers reported that the main difficulty patients faced as part of the intervention was the use of technology; a lack of motivation and skills for self-learning were also reported as difficulties their patients faced. In regard to their own difficulties, the healthcare providers that delivered the intervention reported that it was hard to encourage the patients over the phone to complete all their weekly tasks and sustain healthy habits. Suggestions to improve the delivery of the educational intervention included having video calls (instead of phone calls), having live exercise sessions with the patients, and live educational webinars specific to certain conditions, such as diabetes.

### 3.6. Limited Efficacy Testing

Regarding limited efficacy, the intervention was associated with positive changes regarding exercise self-efficacy, sleep quality, depressive symptoms, and performance of high-intensity levels of physical activity (Table 4). This preliminary study showed that the intervention was able to increase exercise self-efficacy, improve sleep quality, decrease depressive symptoms, and increase the percentage of participants performing high-intensity levels of physical activity. However, disease-related knowledge decreased post-intervention.

## 4. Discussion

Although COVID-19 accelerated the shift to virtual healthcare for those at most risk [24] (including CVD patients), the need to create alternative CR delivery models to increase access to and participation in these programmes was paramount prior to the pandemic. Our study results suggest the potential use of a 12-week educational intervention delivered by WhatsApp in combination with 6 phone calls with healthcare providers as a feasible and acceptable means for delivering cardiovascular health education to promote behaviour change. The patient participants reported overall positive satisfaction with the virtual education programme, mainly due to learning about exercise and their overall health condition, maintaining or improving their health through education, and staying in touch with the CR programme during the pandemic. The healthcare providers that delivered the education also reported overall positive perceptions, mainly related to bi-weekly phone calls and educational content aligned to the needs of their patients. Overall, all 34 patient participants accessed all 12 weekly educational sessions and watched the corresponding video and read parts of the booklet. In addition, most of the patients reported positive changes in their heart health behaviours after the intervention, which was also confirmed by changes in the scores pre- to post-intervention.

Patient education is recommended for patients who have CVDs to increase their knowledge and encourage them to adopt heart-healthy behaviours and promote self-care, which can reduce disease progression and improve clinical outcomes [25,26,27]. The literature shows that structured educational interventions—in a variety of modes and intensities—are effective at improving outcomes [25]. A systematic review by Shi et al. [25] found that few studies were published on virtual educational interventions designed for coronary artery disease patients prior to 2020. In the last few years, the involvement of digitalised educational modalities has been increasing in cardiology [28]; therefore, the availability of a feasible educational intervention is important to contribute to the care of cardiac patients. In addition, patients face multilevel barriers to CR enrolment and participation [29] (including in Brazil [30]), and important barriers could be mitigated by a virtual intervention, such as access barriers (e.g., distance, cost, and transportation).

A lack of motivation and skills for self-learning were reported by the healthcare providers delivering the education as difficulties that their patients faced during the virtual intervention. The providers also reported how hard it was to motivate the patients to sustain healthy habits. Health education is one of the most important elements of health promotion [31]; however, most healthcare providers are not trained to effectively educate patients and their families [32]. Previous studies reported barriers that healthcare providers faced to educating their patients, including a lack of motivation, inappropriate communication skills and conflict, and a lack of coherence in education [32,33,34,35]. In addition, the format of the delivery material and social support are also considered barriers to virtual learning in cardiac patients [36]. Therefore, training healthcare providers on adult learning principles and teaching strategies, as well as delivering education in a format where patients can have peer support (i.e., group video sessions), are advised. Social support is considered a key motivation for virtual learning in CR programmes, in which education is delivered in groups and in real-time [36,37].

Studies have identified multiple barriers to virtual education within the CR context. A recent systematic review of qualitative studies identified the didactic format of the virtual sessions (e.g., educational sessions too long and usually based on PowerPoint slides), problems with technology (e.g., a lack of access to an internet-connected computer and audio-visual and connectivity issues), and a lack of social support (i.e., a lack of interaction in virtual interventions) as barriers [36]. The pandemic has posed significant challenges in the activities related to education, including patient education. As previously mentioned, CR programmes had to quickly transition to virtual models of care after the start of the COVID-19 pandemic [11]; therefore, the identification of virtual interventions that work well for patients, as well as barriers to virtual education, should always be explored to support patient learning.

Although the findings of this study are encouraging, the results should be interpreted with caution. First, the results are limited to the participants of one public CR programme in the south of Brazil; therefore, the generalisability of the findings is limited. It is recommended to test the feasibility of this intervention in other settings, such as rural areas where broadband access is poorer. In addition, other feasibility measures were not tested, including customisation, integration, and extension testing. Second, selection bias toward a highly motivated group might have existed, as the duration of CR participation for most of the patient participants was more than 13 months. Third, due to the nature of the study, causal conclusions cannot be drawn. The results from this study should form the basis for the design of a future randomised controlled trial evaluating the effectiveness of this intervention—including the effects on health parameters, such as blood pressure or cholesterol levels— and also compare the results to a control group of other interventions. Future studies should also focus on implementing the intervention in other patient populations—including those that do not highly adhere to CR—as well as assess the effects in patients grouped by age, gender, and cardiovascular pathologies. Moreover, the content of the educational sessions should be expanded to address the needs of other patient populations, such as those living with diabetes.

## 5. Conclusions

The usage of a 12-week virtual educational programme for behaviour change was demonstrated to be feasible and acceptable for the education of cardiac patients from a low-resource setting in Brazil. The programme showed preliminary effectiveness in increasing exercise self-efficacy, improving sleep quality, decreasing depressive symptoms, and increasing the percentage of participants performing high-intensity levels of physical activity. The virtual programme was also well received by patients and healthcare providers, yet challenges were identified that should be addressed to improve the intervention. Adjustments to the educational content and delivery mode and support to complete the tasks are recommended for future trials.

## Figures and Tables

**Figure 1 ijerph-20-05934-f001:**
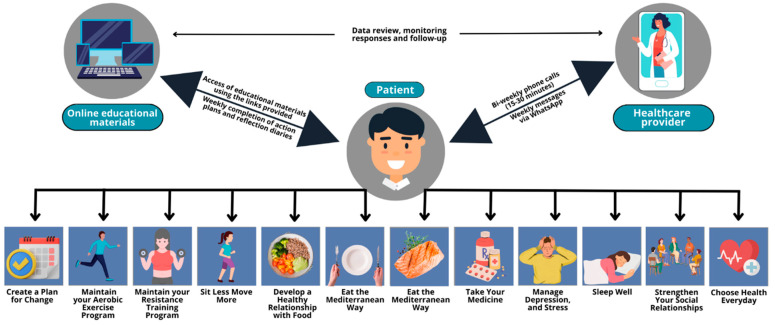
Educational intervention.

**Table 1 ijerph-20-05934-t001:** Feasibility measures: definition, outcomes, data source, and data analysis.

Feasibility Measure	Definition	Outcomes	Data Source	Data Analysis
Acceptability	To what extent the intervention is suitable and satisfying for target individuals	Patients’ satisfaction with the intervention	Satisfaction questionnaire designed for patient participants (6 questions)	Descriptive statistics (Likert-type scale) and qualitative analysis (open-ended questions)
		Healthcare providers’ satisfaction with delivering education virtually to patients	Satisfaction questionnaire designed for healthcare providers delivering the intervention (3 questions)	
Demand	To what extent the programme is likely to be used	Patients: internet access, device to access Internet, use of Internet to search for health information, reasons to participate in the virtual educational programme	Questionnaire about Internet use (4 questions)	Descriptive statistics (frequency)
		Perceived demand reported by patients (usefulness of information and effectiveness of bi-weekly)		
Implementation	To what extent can the intervention be fully implemented as planned	Access and use of virtual education tools (weekly educational materials, completion of weekly action plans and reflection diaries)	Data recorded from the links provided and self-reported by patients	Descriptive statistics (frequency) and qualitative analysis (open-ended questions)
		Healthcare providers’ perceptions about success and failure of the intervention	Semi-structured 1:1 interview with healthcare providers delivering the intervention	
Practicality	To what extent can the intervention be performed by participants using the intended means and resources	Ability of patient participants to carry out intervention activities and maintain healthy habits after end of intervention	Semi-structured 1:1 interview with patient participants	Qualitative analysis (open-ended questions)
		Healthcare providers’ perceptions about factors affecting implementation ease or difficulty	Semi-structured 1:1 interview with healthcare providers delivering the intervention	
Limited efficacy testing	What are the preliminary impacts of the intervention on study variables	Preliminary effects of the intervention on disease-related knowledge, exercise self-efficacy, sleep quality, depressive symptoms, and physical activity level	Psychometric validated questionnaires completed by patient participants pre- and post-intervention	Descriptive statistics (Δ post-pre)

**Table 2 ijerph-20-05934-t002:** Sociodemographic and clinical characteristics of patient participants (n = 34).

Characteristic	n (%)
*Sex*	
Male	22 (64.7)
Female	12 (35.3)
*Marital Status*	
Married	21 (61.8)
Divorced	5 (14.6)
Widow	4 (11.8)
Single	4 (11.8)
*Highest Educational Level*	
High school or lower	20 (58.8)
More than high school	13 (38.3)
No information	1 (2.9)
*Monthly Family Income ^a^*	
5 minimum wages or under	20 (58.8)
More than 5 minimum wages	8 (23.5)
No information	6 (17.6)
*Cardiac Diagnosis/Procedures ^b^*	
Heart failure	22 (64.7)
Acute myocardial infarction	21 (61.8)
Percutaneous coronary intervention	18 (52.9)
Coronary artery disease	18 (52.9)
Coronary artery bypass graft	10 (29.4)
*Cardiovascular Risk Factors ^b^*	
Hypertension	25 (73.5)
Dyslipidaemia	15 (44.1)
Smoking (current or past)	15 (44.1)
Diabetes Type II	12 (35.3)
*Duration of CR Participation*	
Between 6 and 12 months	5 (14.7)
12 months	3 (8.8)
Between 13 and 18 months	20 (58.8)
More than 18 months	6 (17.6)

*^a^* minimum wage in Brazil corresponds to BRL 1045.00 (USD 202.78) in 2020/2021. *^b^* multiple cardiac diagnoses and risk factors could be selected by one participant.

**Table 3 ijerph-20-05934-t003:** Feasibility outcomes for patients (n = 20) and healthcare providers (n = 8).

Feasibility Measure	Outcome	Description	Results ^a^
Acceptability	Satisfaction (patients)	Overall satisfaction with the programme	9.0 (7.4–10.0)/10
		Satisfaction with the educational content	9.0 (8.6–9.5)/10
		Satisfaction with the delivery of education (i.e., via WhatsApp)	9.0 (8.1–10.0)/10
		Satisfaction with the action plans and diaries	9.1 (8.9–10.0)/10
	Satisfaction (healthcare providers)	Overall satisfaction with the programme	9.8 (9.6–10.0)/10
		Satisfaction with the educational content	8.5 (7.3–9.0)/10
		Satisfaction with the delivery of education (i.e., via WhatsApp)	8.1 (7.7–9.0)/10
Demand	Internet use (patients)	Have internet access	18 (90.0)
		Device mostly used to access internet: cell phone	16 (80.0)
		Use of Internet to search for health information	14 (70.0)
	Reasons to participate in the virtual educational programme (patients)	Opportunity to learn about exercise	8 (40.0)
		Maintain or improve health through education	8 (40.0)
		Learn about their overall health condition	5 (25.0)
		Stay in touch with programme during the pandemic	3 (15.0)
	Perceived demand (patients)	Usefulness of information, yes	20 (100.0)
		Effectiveness of bi-weekly calls, yes	18 (90.0)
Implementation ^b^	Access and use of virtual education tools	Use of weekly educational materials (video and booklet)	34 (100.0)
		Creation of weekly learning plans	18 (53.0)
		Creation of reflection diaries	16 (47.0)
Practicality	Ability of patient participants to carry out intervention activities and maintain healthy habits after end of intervention	Heart health behaviours changed after intervention, yes	19 (95.0)

^a^ Results presented as median (IQR) or n (%). ^b^ Results from implementation were collected for 34 patient participants.

**Table 4 ijerph-20-05934-t004:** Pre-, post-intervention scores and Δ post-pre for the measurements used to investigate limited efficacy (n = 34).

	Maximum Score	Pre-Intervention Scores	Post-Intervention Scores	Δ Post-Pre *
Disease-related knowledge	20	8.9 ± 3.2	6.9 ± 4.2	−2.0
Exercise self-efficacy	100	53.9 ± 15.9	56.7 ± 13.7	+2.8
Sleep quality	21	6.4 ± 4.3	5.3 ± 4.4	−1.1
Depressive symptoms	6	1.6 ± 1.9	1.1 ± 1.4	−0.5
High-intensity physical activity level	34 (100%)	6 (40%)	14 (70%)	+30%

*—decrease in the Δ post-pre; + increase in the Δ post-pre.

## Data Availability

Not applicable.

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
