# Peer review of "Feasibility of a Virtual Educational Programme for Behaviour Change in Cardiac Patients from a Low-Resource Setting"

_ijerph, 2023, doi:10.3390/ijerph20115934_

Round 1

Reviewer 1 Report

investigated the feasibility of a virtual educational program for behavior change in cardiovascular rehabilitation patients from a low-resource setting. The authors concluded that the intervention could educate cardiac patients from a low-resource setting. Studies to explore possible ways to deliver patient education remotely are of great importance, especially during the pandemic. Overall, the rationale, method, study design, results, and conclusions were presented in an organized and clear way. The references were appropriate, and the figure and tables look good. The manuscript is very well written. My only suggestion for this manuscript is that the authors may consider discussing more the current limitations of virtual education in the discussion.

Reviewer 2 Report

In general the study is well designed and clearly described.

Nevertheless, from my point of view, the authors should improve some aspects before being published. Further details about my observations and suggestions to the authors can be found in the attached report.  

Reviewer 3 Report

 Feasibility of a virtual educational program for behaviour 2 change in cardiac patients from a low-resource setting

-         peer review

This longitudinal study followed the usage of a 12-week virtual educational program aimed at promoting behaviour change in patients undergoing cardiovascular rehabilitation. A total of 34 patients and 8 healthcare workers were included in the study, which presented an intriguing approach for the potential integration of modern technologies (remote communication via social networks) in cardiovascular rehabilitation.

1.   Introduction

Although the introduction reflects the paper accordingly, the very significance of the CR should be explained more thoroughly.

2.   Methods

Lines 117-118: It would be helpful if the authors could provide an explanation for their choice of the IPAQ-SF questionnaire as an indicator of physical activity, considering the conclusions of the study [22].

3.   Results

Overall, the results require major changes. This includes enhancing the comprehensiveness and precision of the paragraphs and improving consistency across the entire section.

Table 2 needs to be improved:

a.       The order of the sub-divisions is confusing

b.       It is unclear why is the COPD listed under the Cardiac Diagnosis/Procedures

c.       Hypertension is listed under the risk factors while it can also be argued it is a cardiac diagnosis.

Table 3 needs to be highly improved. It is confusing to some extend and lacks consistency.

a.       What was the reason of only 20 patients completing the feasibility questionnaire?

b.      It appears that in some cases, the Authors have reported the results for all 34 participants (Implementation).

c.       Numeric variables should be presented as medians with the IQR or some other variability indicator since the variables are discrete, not continuous, e.g., 9.0 ±2.4/10 is confusing since the score cannot go over 10.

d.      The connection between Table 2 and 3 is unclear. Healthcare providers’ perception was not reported in the Table 3 regarding the Implementation and Practicality.

e.       In conclusion, Table 3 has to be structured better.

Line 172, “mean of 18 patients” – this should be rephrased since it is not the mean that was reported in the table. This also goes for the Line 175.

Table 4 needs to be improved:

Authors should add the short interpretation for the scores of the questionnaires they have used. + and – signs in front of the Δ post- pre values mean different things for the different scores, this should be addressed.

4.   Discussion

The discussion has to be improved. The results from this study have to be more thoroughly discussed and compared with the available literature.

Study limitations are appropriate and well phased. 

Reviewer 4 Report

This manuscript presents a feasibility study of a virtual educational program for behavior change in cardiac patients from a low-resource setting. It included a total of 42 participants (34 patients and 8 healthcare workers). Feasibility measures included acceptance, demand, implementation, practicality and limited effectiveness testing, with a lack of customization, integration and extension testing. This should be emphasized in the text. The results are well interpreted, and the shortcomings are shown in the discussion. The conclusion is appropriate and based on the results obtained.
